# Culture-Dependent and Metabarcoding Characterization of the Sugar Beet (*Beta vulgaris* L.) Microbiome for High-Yield Isolation of Bacteria with Plant Growth-Promoting Traits

**DOI:** 10.3390/microorganisms11061538

**Published:** 2023-06-09

**Authors:** Tamara Krstić Tomić, Iva Atanasković, Ivan Nikolić, Nataša Joković, Tatjana Stević, Slaviša Stanković, Tanja Berić, Jelena Lozo

**Affiliations:** 1University of Belgrade, Faculty of Biology, 11000 Belgrade, Serbia; m3004_2020@stud.bio.bg.ac.rs (T.K.T.); iva.atanaskovic@bio.bg.ac.rs (I.A.); ivan.nikolic@bio.bg.ac.rs (I.N.); slavisas@bio.bg.ac.rs (S.S.); tanjab@bio.bg.ac.rs (T.B.); 2University of Belgrade, Faculty of Biology, Centre for Biological Control and Plant, Growth Promotion, 11000 Belgrade, Serbia; 3Faculty of Sciences and Mathematics, University of Niš, 18000 Niš, Serbia; natasa.jokovic@pmf.edu.rs; 4Institute for Medicinal Plants Research “Dr. Josif Pančić”, 11000 Belgrade, Serbia; tanjasomosa@gmail.com

**Keywords:** sugar beet, plant microbiome, 16S rRNA metabarcoding, plant-based media, seasonal changes

## Abstract

The diversity of plant-associated bacteria is vast and can be determined by 16S rRNA gene metabarcoding. Fewer of them have plant-beneficial properties. To harness their benefits for plants, we must isolate them. This study aimed to check whether 16S rRNA gene metabarcoding has predictive power in identifying the majority of known bacteria with plant-beneficial traits that can be isolated from the sugar beet (*Beta vulgaris* L.) microbiome. Rhizosphere and phyllosphere samples collected during one season at different stages of plant development were analyzed. Bacteria were isolated on rich unselective media and plant-based media enriched with sugar beet leaves or rhizosphere extracts. The isolates were identified by sequencing the 16S rRNA gene and tested in vitro for their plant-beneficial properties (stimulation of germination; exopolysaccharide, siderophore, and HCN production; phosphate solubilization; and activity against sugar beet pathogens). The highest number of co-occurring beneficial traits was eight, found in isolates of five species: *Acinetobacter calcoaceticus*, *Bacillus australimaris*, *B. pumilus*, *Enterobacter ludwiigi*, and *Pantoea ananatis*. These species were not detected by metabarcoding and have not previously been described as plant-beneficial inhabitants of sugar beets. Thus, our findings point out the necessity of a culture-dependent microbiome analysis and advocate for low-nutrient plant-based media for high-yield isolation of plant-beneficial taxa with multiple beneficial traits. A culture-dependent and -independent approach is required for community diversity assessment. Still, isolation on plant-based media is the best approach to select isolates for potential use as biofertilizers and biopesticides in sugar beet cultivation.

## 1. Introduction

Agriculture has an enormous environmental footprint and majorly contributes to climate change. Sustainable agricultural practices aim to improve soil fertility and crop production while protecting the environment [1]. One aspect of this is the replacement of synthetic fertilizers and pesticides with biofertilizers and biocontrol agents. These products contain living microorganisms and, when applied to plants or soil, promote plant growth and prevent infection of the host plant by pathogens. Such plant-beneficial microorganisms are found in the rhizosphere and phyllosphere of healthy plants [2]. Therefore, research into the microbiome of crops is an important step in developing biofertilizers and biocontrol agents. Metabarcoding, particularly the analysis of the hypervariable regions of the 16S rRNA gene, is a culture-independent technique commonly used in microbiome studies [3]. However, to test causality, e.g., whether a microbe promotes plant growth and applies it in the field, it needs to be cultivated. To minimize the gap between gene-detected and cultivable bacterial diversity, developing novel culturing strategies to increase the number of species isolated from the plant-associated microbiota is imperative. The necessity of cultured bacteria for many applications led to the renaissance of enrichment methods. In this sense, defining plant-based culture media containing host plant extracts, or media enriched with soil extracts, has proven particularly useful for isolating plant-beneficial microbes [4,5]. Plant-based media contain nutrients present in plant exudates and mimic conditions on the plant surface or inside tissues. Therefore, these media enhance the isolation of microbes that are closely associated with the host plant, many of which are important for host growth.

The sugar beet (*Beta vulgaris* L.) is the most important crop for the sugar industry in temperate regions because of its high biomass production, even under harsh environmental conditions [6]. In Europe, sugar production from the sugar beet was about 200,896,204 tons in 2021 [7]. Its resilience under stress conditions could be due to beneficial microorganisms in the sugar beet microbiome. While the rhizobiome of the sugar beet has been previously characterized [6], little is known about the phyllobiome and the plant-beneficial microbes that inhabit this plant niche. Several species have been described as plant growth-promoting bacteria (PGPB) of the sugar beet, including *Acinetobacter*, *Bacillus*, *Pseudomonas*, *Paenibacillus*, *Sphingomonas*, *Stenotrophomonas*, and *Nocardioides* spp. [8,9]. These bacteria exhibit various plant-beneficial traits, such as germination and growth stimulation, production of plant hormones, and siderophores. However, considering the overall phylogenetic diversity of the sugar beet microbiome [6,8], PGPBs associated with this plant may be more numerous than we currently know.

Previous attempts to isolate PGPBs from the sugar beet relied on commercial, nutrient-rich media with very high nutrient content, which does not really reflect nutritional conditions in the rhizosphere, phyllosphere, and sugar beet tissues and may not successfully isolate many plant-beneficial bacteria inhabiting plants. Therefore, using plant-based media could expand the list of plant-beneficial microorganisms in the sugar beet microbiome. There is ample evidence that plant-based media can optimize the recovery of bacteria from plant tissue. Compared to commercially available synthetic culture media (e.g., nutrient agar), media containing manure homogenates, crude juices, saps, and powders from cacti (*Opuntia ficus* Indica) and succulents (*Aloe vera* and *A. arborescens*) were more efficient in isolating rhizobacteria from the ecto- and endo-rhizospheres of the host plants tested [10]. Another strategy is to change the preparation of the agar media and autoclave the gelling agent separately. Combining this approach with prolonged cultivation of agar plates was sufficient for the isolation of phylogenetically novel bacteria from forest soil and pond sediment samples [11].

Furthermore, the choice of gelling agents, such as agarose or phytagel, may impact the isolation of rhizobacteria on plant-based media [12]. In addition, using plant extracts, the enrichment of media with rhizosphere extracts has also been shown to be effective for isolating rhizobacteria. The addition of soil methanol extracts has been used to efficiently isolate previously uncultivated bacteria from the rhizosphere of *Robinia pseudoacacia* [5]. Therefore, the detectability of plant-associated microbes can be improved by combining these approaches.

Bacterial community composition determined by 16S rRNA gene metabarcoding (usually amplifying the V3–V4 hypervariable region) becomes the standard output in different research. Despite its proven superiority over culture-dependent methods, we hypothesized that the isolation of bacteria onto specialized media could prove to be equally sensitive or even more sensitive when the goal is detecting and isolating bacteria with special characteristics. We used different media, aiming at isolating a high yield of bacteria from the rhizosphere and phyllosphere of the sugar beet and subsequently tested for various plant-beneficial traits: germination stimulation; exopolysaccharide, siderophore, and HCN production; phosphate solubilization; and biocontrol activity against common sugar beet pathogens associated with severe yield losses (*Pseudomonas syringae* pv. *aptata*, the causal agent of bacterial leaf spot disease [13], *Fusarium oxysporum*, the causal agent of root rot diseases [14], and *Rhizoctonia solani*, the causal agent of *Rhizoctonia* root and crown rot diseases [15]). The goal was to compare the lists of taxa obtained by both methods and to determine whether 16S metabarcoding gives insight into the presence of bacteria with several co-occurring plant-beneficial characteristics. 

## 2. Materials and Methods

### 2.1. Plant Material and Sampling

Sugar beets used for this study were grown at the experimental field (44°56′2088″ N and 20°43′2145″ E) of PSS “Tamiš” Institute, Agriculture Extension Service Province of Vojvodina, Pančevo, Serbia. Sampling was done three times in 2021, 3 (May), 5 (July), and 8 (October) months after sowing. Samples were from the sugar beet variety Eduarda (KWS, Einbeck, Germany), the most commonly grown variety in Serbia. Three independent replicates were taken from the phyllosphere and rhizosphere of three different plants for all three seasonal samples. The plant material was collected in sterile plastic bags and transported in a cool box to the laboratory for further processing.

### 2.2. DNA Isolation for 16S rRNA Metabarcoding

The diversity of bacteria in the rhizosphere and phyllosphere was analyzed using a metabarcoding sequencing approach. Rhizosphere sample was prepared by treating plant material in an ultrasonic bath (Bandelin conorex, Berlin, Germany) for 10 min at 35 kHz and then collecting soil attached to the root and material scraped from the root surface to obtain a sample containing epiphyte and endophyte bacteria. DNA extraction for the rhizosphere samples (250 mg per each sample) was performed using Zymo-BIOMICS^TM^ DNA Miniprep Kit (Zymo Research, Irvine, CA, USA) according to the instructions of the manufacturer. Phyllosphere (one leaf per replicate) was cut into small pieces and shaken on a rotary shaker (150 rpm) at 22 °C for 15 min with 1× phosphate-buffered saline (PBS, Sigma Aldrich, Gillingham, UK) for each sample independently. After preparing plant material, 100 mL of each sample was filtered with 0.22 µm Isopore^TM^ membrane filters (Merck Millipore Ltd., Carrigtwohill, Ireland). Total DNA was extracted from filters of all independent samples and replicates using the Zymo-BIOMICS™ DNA Miniprep Kit (Zymo Research, Irvine, CA, USA) according to the manufacturer’s instructions, and DNA concentration was determined using the Nanophotometer spectrophotometer (IMPLEN, Munich, Germany). DNA samples dissolved in DNase/RNase-free water were commercially sequenced by Novogene Bioinformatics Technology Co., Ltd. (Cambridge, UK).

### 2.3. DNA Sequencing and Analysis

The amplicon was sequenced on a paired-end Illumina platform NovaSeq PE250 to generate 250 bp paired-end raw reads (Raw PE), which were then merged using FLASH (V1.2.7) [16], filtered under specific conditions [17] using Qiime (V1.7.0) quality-controlled process [18] to obtain Clean Tags. The chimeric sequences in the Clean Tags were detected using the UCHIME algorithm [19] and removed [20] to obtain the Effective Tags, which could be used for subsequent analysis. The 16S rRNA gene-specific sequences targeting the V3 and V4 regions were used in this study. Sequence analysis was performed with Uparse software (v7.0.1090) using all Effective Tags [21]. Sequences with ≥97% similarity were assigned to the same OTUs. The representative sequence for each OTU was screened for further annotation. For each representative sequence, the Qiime [22] in the Mothur method was performed against the SSUrRNA database from the SIL-VA138 Database [23] for annotation of species in each taxonomic rank [24]. For phylogenetic relationship of all OTU representative sequences, the MUSCLE [25] was used. OTU abundance information was normalized using a standard of the sequence number corresponding to the sample with the fewest sequences.

The following alpha diversity and beta diversity analyses were all performed based on these output-normalized data. Alpha diversity is used to analyze the biodiversity complexity of a sample using 6 indices, including Observed-species, Chao1, Shannon, Simpson, ACE, and Good-coverage. All these indices were calculated in our samples using Qiime (Version 1.7.0) and displayed using R software (Version 2.15.3). The Wilcoxon rank-sum test was used for testing significance among all groups of samples and the *p*-values < 0.05 were considered significant. Beta diversity analysis was used to assess differences in species complexity among samples. Beta diversity was calculated for both weighted and unweighted UniFrac using Qiime software (Version 1.7.0). Cluster analysis was preceded by principal component analysis (PCA) applied to reduce the dimensions of the original variables using the FactoMineR package and the ggplot2 package in R software (Version 2.15.3). Principal Coordinate Analysis (PCoA) was performed to obtain the principal coordinates and visualize complex multidimensional data. Unweighted Pair-group Method with Arithmetic Means (UPGMA) clustering was performed as a hierarchical clustering method to interpret the distance matrix using the average linkage. It was performed using Qiime software (Version 1.7.0). Community differences analysis that includes Anosim, MRPP, and adonis was performed using R software (V.2.15.3) (Vegan package: anosim function, mrpp function, and adonis function). AMOVA was calculated by mothur using the amova function. The confidence degree for all community difference analyses is represented by a *p*-value, whose value less than 0.05 suggests statistical significance. T-test was performed to determine species with significant variation between groups (*p*-value < 0.05) at various taxon ranks. T-test and drawing were performed using R software (Version 2.15.3).

The data have been entered into the GenBank database (NCBI) under BioProject Acc. No. PRJNA939425.

### 2.4. Isolation and Characterization of Bacteria from Plant Samples

Bacteria were isolated from the phyllosphere and rhizosphere of sugar beets. The leaves were used for the phyllosphere samples, while the rhizosphere samples included the exorhizosphere, rhizoplane, endorhizosphere, and the soil attached to the roots. The samples of leaves were prepared in two different ways to isolate endophytic and epiphytic bacteria. For endophyte isolation, we followed the procedure of de Oliveira Costa et al. [26] with modifications. Briefly, 10 g of leaves/roots were sterilized with 70% ethanol for 30 s. Ethanol was discarded, and 2.5% sodium hypochlorite was added and shaken periodically for 8 min. After washing the samples with distilled water five times, they were transferred to a mortar, and 5 mL of saline solution (0.9% NaCl, *v*/*v*) was added. The samples were crushed with a pestle until a homogenous mixture was obtained. Dilutions were prepared from the sample and 100 μL of the appropriate dilution was transferred to a suitable substrate. To isolate epiphytes, 10 mL of saline solution was mixed with 10 g of the plant material, shaken for 3 h at 120 rpm, and the same procedure as for endophytes was repeated. Initial dilution of the rhizosphere sample was prepared by mixing 20 g of the sample in 200 mL of sterile saline. The mixture was stirred for 30 min and then allowed to stand at room temperature for 15 min to separate suspension and sediment before use. For all samples, higher serial dilutions (10^−4^, 10^−5^, and 10^−6^) prepared in sterile saline were applied to solid media. Plates were incubated at 25 °C for 4 weeks. After every 7 days, morphologically distinguishable colonies were collected and streaked onto appropriate sterile agar plates. The media used for phyllosphere samples were leaf extract agar (LEa), leaf extract phytagel (LEph), PS, and nutrient agar (NA) (Torlak, Belgrade, Serbia), and for rhizosphere samples, root extract agar (REa), root extract phytagel (REph), NA, PS, intensive soil extract medium 1 (ISEM1), ISEM2, methanol soil extract medium (MSEM), water soil extract medium (WSEM), 0.1 tryptic soy agar (TSA) (TH Media, Titan Biotech, Ltd., Delhi, India), 0.01 TSA, water yeast agar (WY), soil extract agar (SEa), and soil extract phytagel (SEph) (see Table 1 for details).

One week after the last transfer of colonies to the appropriate medium, they were all plated on sterile TSA medium and incubated at 25 °C until colony appearance. However, isolates that could not grow on the TSA medium or grown poorly were propagated on the medium where they were initially isolated. The next step was Gram and endospore staining involving the growth pattern of colonies on Simmons citrate, Bile esculin, and Endo agar media (Torlak, Belgrade, Serbia). The production of exoenzymes such as amylase, protease, gelatinase, cellulase, and mannanase [27] and catalase assay was also tested. Based on these features, the bacterial isolates were preliminarily characterized, and 531 isolates were selected for further analysis.
microorganisms-11-01538-t001_Table 1Table 1Composition of the media used for the culture-dependent method of bacterial isolation.MediumCompositionReference0.1 TSATSB 3 g/L, agar 15 g/L[28]0.01 TSATSB 0.3 g/L, agar 15 g/LWYNaCl 5 g/L, yeast extract 0.05 g/L, agar 20 g/LPSsolution A: (NH_4_)_2_SO_4_ 2.27 mM, MgSO_4_ 0.2 mM, CaCl_2_ 45 μM, agar 15 g/Lsolution B: KH_2_PO_4_ 10 mM, Na_2_HPO_4×_12H_2_O 10 mMsolution C: peptone 0.1 g/L, yeast extract 0.1 g/L, glucose 0.1 g/L[11]LEaleaf extract 50 mL/L, agar 20 g/L[4]LEphleaf extract 50 mL/L, phytagel 20 g/LREaroot extract 50 mL/L, agar 20 g/LREphroot extract 50 mL/L, phytagel 20 g/LSEaglucose 1 g/L, peptone 1 g/L, yeast extract 1 g/L, K_2_HPO_4_ 1 g/L, soil extract 400 mL/L, agar 15 g/L[12]SEphglucose 1 g/L, peptone 1 g/L, yeast extract 1 g/L, K_2_HPO_4_ 1 g/L, soil extract 400 mL/L, phytagel 20 g/LISEM1I mineral salts: K_2_HPO_4_ 0.23 g/L, MgSO_4×_7H_2_O 0.23 g/L, NH_4_NO_3_ 0.33 g/L, NaHCO_3_ 0.25 g/L, agar 15 g/LII amino acids: d-valine 5 mg, d-methionine 5 mg, d-leucine 5 mg, d-phenylalanine 5 mg, d-threonine 5 mg, and d-tryptophan 5 mgIII vitamin B: 1 mL vitamin stock solution containing 50 mg each thiamine hydrochloride, riboflavin, niacin, pyridoxine HCl, inositol, calcium pantothenate, and β-aminobenzoic acid and 25 mg biotin in 100 mL distilled waterIII selenite–tungstate solution: 2 mL/L compositions in 1 L of distilled water: NaOH 0.5 g, Na_2_SeO_3_·5H_2_O 3 mg, Na_2_WO_4×_2H_2_O 4 mgIV trace elements: 2 mL/L HCl [25%, *v*/*v*] 10 mL, FeCl_2_·4H_2_O 1.5 g, ZnCl_2_ 70 mg, MnCl_2×_4H_2_O 100 mg, H_3_BO_3_ 6 mg, CoCl_2×_6H_2_O 190 mg, CuCl_2×_2H_2_O 2 mg, NiCl_2×_6H_2_O 24 mg, Na_2_MoO_4×_2H_2_O 36 mg in a final volume of 1 literV methanol soil extract 200 mL/L[5]ISEM2All is the same as in ISEM1 but V is different where water soil extract 200 mL/L was used.MSEMI mineral salts as in ISME with methanol soil extract 200 mL/LThis workWSEMI mineral salts as in ISME with water soil extract 200 mL/L


### 2.5. Bacteria Identification to Species Level

Genomic DNA from selected bacterial isolates was isolated as previously described [29]. Sequencing of the 16S rRNA gene amplicon was performed for molecular identification of bacterial isolates using the universal primers UN1-16SF (5′-GAGAGTTT-GATCCTGGC-3′) and UN1-16SR (5′-AGGAGGTGATCCAGCCG-3′). In addition, a PCR reaction with primers for the *tuf* genes tufGPF (5’-ACGTTGACTGCCCAGGACAC-3′) and tufGPR (5′-GATACCAGTTACGTCAGTTGTACGGA-3′) was performed to identify species of the genus *Bacillus*. Amplification of the *gyrB* gene, which encodes the B subunit of DNA gyrase, was analyzed for *Pseudomonas* spp. using GyrB-F (5′-MGGCGGYAAGTTCGATGACAAYTC-3′) and GyrB-R (5′-TRATBKCAGTCARACCTT-CRCGSGC-3′) primers [30]. PCR amplifications were performed using DreamTaq TM Green PCR Master Mix 2× (Thermo Fischer Scientific, Waltham, MA, USA). PCR products were purified using a PCR purification kit (Thermo Fischer Scientific, Waltham, MA, USA) according to the manufacturer’s instructions. The sequences of the PCR products were determined by Eurofins (Koeln, Germany). Data analysis was performed using the program BLAST (Basic Local Alignment Search Tool).

### 2.6. Characterization of Plant-Beneficial Traits

Plant growth-promoting (PGP) activities tested in this study included siderophore production, HCN production, phosphate solubilization, exopolysaccharide production, and antibacterial, antifungal, and seed germination activities. Chrome azurol S (CAS) agar was used to detect siderophore production [31]. To test HCN production, a medium containing nutrient agar and 4.4 g/L glycine was used. The isolates were streaked onto a medium and a Whatman filter paper soaked with 2% sodium carbonate in 0.5% picric acid solution was placed on the top of the Petri dish. The plates were sealed with parafilm and incubated at 30 °C for 4 days. The production of HCN could be recognized by the fact that the color of the filter paper changed from yellow to orange, red, or brown. The bacterial isolates were analyzed for phosphate solubilization using Pikovaskaya medium, onto which the cultures were spotted and incubated at 30 °C for 14 days. A clear zone around the bacterial growth indicated a positive result for phosphate solubilization. Exopolysaccharide production was determined with a toothpick, and stretched colonies were designed as producers. All experiments were performed twice independently with three replicates.

The antibacterial activity of the isolates was tested against three indicator strains, pathogenic sugar beet bacteria *Pseudomonas syringe* pv. *aptata* CFBR2473, P16, and P21. The indicator strains were inoculated onto LB agar plates and 10 µL of an overnight culture of the isolates was spotted on the inoculated agar plates. The plates were incubated at 30 °C for 24 h and clear transparent zones around the colonies of the tested isolates indicated antibacterial activity. Initial screening of the antifungal activity of isolates against the two pathogenic fungi, *Fusarium oxysporum* and *Rhizoctonia solani*, was performed qualitatively in the dual assay. The absence/presence of the inhibition zone was recorded compared to the control with grown fungi only. The bacterial isolates were inoculated on potato dextrose agar PDA (HiMedia Laboratories Pvt Limited, Mumbai, India) about 2.5 cm away from a 7-day-old fungal mycelial plug, and incubated at 25 °C for 7 days. All experiments were repeated twice independently, with three replicates for each bacterium/fungus.

The effect of the isolates on the germination of sugar beet seeds was determined as follows. Sugar beet seeds were sterilized with 70% ethanol for 1 min and in 3% sodium hypochlorite for 15 min and then washed three times in sterile water. For this experiment, pure bacterial cultures (bacteria selected from previous tests) were grown in LB at 30 °C and diluted to a final concentration of 10^8^ colony-forming units (CFU)/mL. The surface-sterilized seeds were immersed in the appropriate bacterial suspension for 1 h, air-dried, and placed in the prepared plastic containers. The plastic containers were sterilized with a 20% sodium hypochlorite solution, three layers of sterile paper moistened with sterile distilled water were placed inside, and the seeds were placed on the surface and incubated for 3 weeks in the dark in the growth chamber at 18 °C. The seeds were regularly irrigated with the same amount of distilled water (10 mL per plastic container), and the place of the plastic trays was changed each day. Subsequent treatments with five replicates with 10 seeds per replicate were performed in two individual experiments. As a control, the prepared seeds were immersed in LB without bacterial inoculation. The effects of bacterial treatment on seed germination were calculated.

## 3. Results

### 3.1. Seasonal Analysis of the Sugar Beet Rhizobiome and Phyllobiome by 16S rRNA Gene Metabarcoding

Samples of the rhizosphere and phyllosphere of the sugar beet were taken after 3, 5, and 8 months of growth in May, July, and October. These samples were analyzed by 16S rRNA gene metabarcoding (the summarizations obtained in each step of data processing are shown in Appendix A). The relative abundance of the detected taxa was determined for the phylum, genus, and species level (Figure 1).

The number of species observed in the phyllosphere was lower than in the rhizosphere in all sampling months, with a decrease in the number of species in both sampling sites from May to October. No statistically significant differences exist in the alpha diversity parameters for all samples (Appendix A). The beta diversity analysis was performed to determine differences between samples. The weighted UniFrac and unweighted UniFrac distances were used to measure the dissimilarity coefficient between pairwise samples. The heat map of weighted UniFrac and unweighted UniFrac distances show that the diversity of bacterial OTUs in the sugar beet rhizosphere differed from the phyllosphere (Appendix A). In addition, PCA plots based on weighted UniFrac distances confirm the clear separation of samples from the rhizosphere and phyllosphere (Appendix A).

Actinobacteriota and Proteobacteria are the two most abundant phyla in all samples. The phyla shown in Figure 1A account for 86%, 98%, and 97% of all phyla detected in the rhizosphere samples for May, July, and October, respectively. As in the rhizosphere, Chloroflexi, Acidobacteriota, and Firmicutes were the most abundant in the phyllosphere (Figure 1B) (>2%). Methylomirabilota, Nitrospirota, and Verrucomicrobiota were detected in both the rhizosphere and the phyllosphere, but did not exceed the 0.5% limit in the phyllosphere. Interestingly, Deferribacterota was detected with high frequency only in the phyllosphere, and only in July. No statistically significant difference in the abundance of OTUs was detected between sampling months in the rhizosphere and the phyllosphere. Still, there are significant differences between the rhizosphere and the phyllosphere for each month (Appendix A). In May, Firmicutes, Crenarchaeota, and Bcatreoidota were more abundant in the rhizosphere. In July, the difference between the two sampling sites was greatest for the Crenarchaeota, which was more abundant in the rhizosphere. Finally, in October, Nitrospirota was more abundant in the rhizosphere than in the phyllosphere.

In the rhizosphere, Nitrososphaeraceae is the most common Archaea genus of phylum Crenarchaeota, and *Bacillus* is the most common bacterial genus of Firmicutes. The most abundant genus in the phyllosphere was *Lactobacillus*, but its abundance decreased drastically in July. Representation of several genera significantly differed between the rhizosphere and phyllosphere in each sampling month (Appendix A). *Nitrososphaeraceae* and *Bacillus* were abundant in both samples but were more prevalent in the rhizosphere in all sampling months. *Pseudomonas* was more abundant in the phyllosphere than in the rhizosphere, but only in October. Apart from this, all other observed differences resulted from certain genera being more abundant in the rhizosphere.

In the rhizosphere, there were 13 species with an abundance of more than 0.2%. Bacillus niacin and *Bacillus simplex* were the two most abundant species for all seasons. Several species were only detected in October: *Bacteroides vulgatus*, *Lactobacillus salivarius*, *Roseburia intestinalis*, and *Streptococcus salivarius*. In the phyllosphere, there were 11 species with an abundance of more than 0.2%. The most abundant species in May and October was *Komagataeibacter saccharivorans*, but it had a low frequency in July. *Bradyrhizobium elkanii* had the highest abundance at all three sampling points. *Lactobacillus johnsonii* and *Lactobacillus murinus* were particularly abundant in July but were not detected in May and October. *Agromyces ramosus*, *Arthrobacter crystallopoietes*, *Bacillus niacin*, *Bacillus simplex*, *Bacteroides vulgatus*, and *Bradyrhizobium elkanii* were detected in both the phyllosphere and the rhizosphere.

### 3.2. Isolation and Characterization of Plant-Beneficial Bacteria from the Sugar Beet Microbiome

Samples of the rhizosphere and phyllosphere of the sugar beet, analyzed by metabarcoding as described in the previous section, were also used to isolate bacteria on different growth media. These media contained different carbon sources, mineral salts, or gelling agents, and some were enriched with leaf, root, or soil extracts (Table 1). Morphologically distinct colonies obtained on each medium were further characterized by biochemical tests (Appendix A) and 16S rRNA gene sequencing. In this way, a total of 403 distinct isolates were obtained from the rhizosphere (Figure 2A) and 128 from the phyllosphere (Figure 2B) of the sugar beet. The highest number of isolates from the rhizosphere (60) was obtained on REa, a medium enriched with the sugar beet root extract. The lowest number of isolates (15) was obtained on WY, a medium with a yeast extract as a carbon source. The highest number of phyllosphere isolates (50) was obtained on NA, a commercial culture medium, and the lowest (20) on LEph, a medium with the leaf extract and phytagel.

All isolates were tested for 10 plant-beneficial traits, Appendix A: germination stimulation; exopolysaccharide, siderophore, and HCN production; phosphate solubilization; and activity against sugar beet pathogens (*Pseudomonas syringae* pv. *aptata* P16, P21, and CFBR2437; *Fusarium oxysporum*; *Rhizoctonia solani*). The number of traits for which each isolate was positive was counted and all isolates were grouped based on the number of co-occurring traits. Furthermore, the frequency of isolates with a certain number of co-occurring traits was calculated for all media (Figure 2C,D). In this way, the medium most suitable for isolating strains from the sugar beet microbiome with plant-beneficial properties was determined. The highest number of co-occurring traits detected in an isolate was eight. They were only detected in the rhizosphere and were mainly isolated on the 0.01 TSA medium. In the phyllosphere, the highest number of co-occurring traits was seven, and the highest frequency of such isolates was obtained on the medium Lea, containing the leaf extract of the sugar beet. As for the month of isolation, the highest frequency of co-occurring features was observed for July (Figure 2E,F) for both rhizosphere and phyllosphere samples, while the lowest frequency was obtained in October.

### 3.3. Comparison of Culture-Dependent and -Independent Approaches in the Detection of Plant-Beneficial Isolates

In this study, all media tested were sufficient for isolating rhizosphere isolates with at least four plant-beneficial traits (Figure 2C,D). Therefore, we focused on isolates that shared five or more properties. These isolates were grouped according to the month of sampling (Figure 3). In May (Figure 3A), the isolate *Acinetobacter calcoaceticus* has the highest number of co-occurring traits (8 out of 10) and was isolated from the rhizosphere on the 0.01 TSA medium. This isolate was active against all sugar beet pathogens tested. Additionally, it stimulated germination, solubilized phosphate, and produced siderophores. The highest number of isolates with eight co-occurring traits is identified in July (Figure 3B)—29 with five or more traits and 3 isolates with eight traits. The ones with eight traits were identified as *Bacillus australimaris* isolated on SEph media and two isolates of *Enterobacter ludwigii* were obtained on 0.01 TSA and SEph media. They were active against all sugar beet pathogens tested, stimulated germination, and produced siderophores. In July, the most common species with five or more plant-beneficial traits was *Stenotrophomonas maltophilia*, isolated on different media and from both the rhizosphere and phyllosphere. From October samples, two strains with eight co-occurring traits (Figure 3C), *Pantoea ananatis* and *B. pumilus*, on the MSEM and SEph mediums, respectively, are indicated. The rest of the plant-beneficial isolates from this sample were identified as *B. pumilus* and *B. safensis*. These isolates were mainly obtained on the REa medium from the sugar beet rhizosphere.

To compare culture-dependent and -independent methods in detecting plant-beneficial strains associated with the sugar beet, we searched for plant-beneficial species listed in Figure 3 in the metabarcoding dataset. The only species with more than five plant-beneficial traits detected both on culture media and by metabarcoding was *B. thuringiensis* (ML15, Figure 3A). This isolate was recovered from the rhizosphere in May on PS media and was detected by metabarcoding in both the rhizosphere and phyllosphere, with a relative abundance of 0.07%. All other plant-beneficial species isolated on different culture media (Figure 3) are not represented among the 179 species detected by metabarcoding. Therefore, the genus level was analyzed in more detail. In Figure 4, each genus from which a plant-beneficial bacterium (Figure 3) was isolated was compared with the month of sampling, the medium on which it was detected, and their relative abundance as determined by metabarcoding (Figure 4).

In the rhizosphere (Figure 4A), the highest number of species with co-occurring plant-beneficial traits is found for the 0.01 TSA medium. These isolates belong to the genera *Acinetobacter*, *Stenotrophomonas*, and *Enterobacter*. All three genera have a low frequency in the metabarcoding dataset, with *Enterobacter* not even being found among the 512 genera detected by metabarcoding. Other rhizosphere genera detected only by a culture-dependent method are *Brucella*, isolated on the media PS and NA, *Paenarthrobacter*, isolated on the medium REph, and *Peribacillus*, isolated on SEa. *Bacillus* is the plant-beneficial genus detected by both methods—it was isolated on 0.1 TSA, PS, ISEM2, REa, MSEM, SEa, and SEph media. It had a relatively high abundance as determined by metabarcoding (>4%). In the phyllosphere (Figure 4B), *Brucella* was the only genus not detected by metabarcoding but isolated on the LEa medium. All other plant-beneficial genera were detected by both methods. *Pseudomonas*, isolated on the medium LEa, contained several plant-beneficial species and was also detected by metabarcoding with a relatively high abundance (>2%). On the other hand, *Stenotrophomonas*, which had the highest number of plant-beneficial species isolated from the phyllosphere on all media tested, was detected by metabarcoding at a very low abundance (<0.5%).

## 4. Discussion

For some time now, metabarcoding of the 16S rRNA gene is a standard tool in microbial ecology used to study the diversity of microbial communities in different habitats, including plants [32,33]. We wondered if that dataset alone would be informative enough to predict the most known plant-associated bacteria with beneficial traits. We compared the lists of taxa obtained by 16S rRNA metabarcoding and the list of isolates obtained by culturing samples on different growth media. Results showed that most plant-beneficial bacteria detected on various plant-based media could not be detected by metabarcoding. The microbiome of the sugar beet has previously been analyzed using metabarcoding from various aspects: seasonal shifts in the lateral root microbiome [6], spatiotemporal changes in endophytic bacterial diversity [34], the rhizobiome of the sugar beet under different fertilizer systems [35], the leaf bacteriome in relation to the susceptibility of the sugar beet to beet curly top virus [36], and Cercospora leaf spot disease [37]. In this study, the microbiome of the sugar beet was investigated in two plant parts, the root and the leaf, and compared in different plant development stages (May, July, and October). In parallel, bacteria were isolated on different plant-based and enriched media, identified by sequencing the 16S rRNA gene, and analyzed for plant-beneficial traits. Therefore, this dataset allowed the comparison of culture-dependent and -independent techniques in detecting plant-beneficial microorganisms associated with the sugar beet.

Metabarcoding revealed no significant variation in alpha diversity parameters across seasons, but the number of OTUs decreased from May to October. Similar observations were made by other researchers, where the alpha diversity of the rhizobiome increased between June and August, saturated in August, and decreased in September [6]. The lower species diversity in the phyllosphere than in the rhizosphere of the sugar beet [38], but also in plants, in general, has been noted previously. This highlights that the phyllosphere is a highly selective habitat for microorganisms susceptible to seasonal diversity fluctuations due to weather changes. For example, the average temperature in Serbia in July 2022 was higher (25 °C) than in May (18 °C) and October (15 °C), which could explain the drastic change in the relative abundance of the most dominant taxa observed in our dataset for July. As in previous studies [6,35,39], Actinobacteriota and Proteobacteria were the most abundant rhizospheric phyla in all seasons. A high abundance of *Bacillus*, a genus rich in PGPB [40], was observed at lower taxonomic levels. Similarly, Ikeda et al. [6] observed a high abundance of *Bacillus*, but only in the early growth stages (May and/or July). Regarding species, *B. niacin* and *B. simplex* were present in large numbers in the rhizosphere at all times of the year. *B. niacin* can use nicotinic acid, a compound present in the root exudate of the sugar beet [41], as its sole source of C and N, and *B. simplex* is a common soil inhabitant that can promote the growth of soybeans and corn [42]. It is unknown whether these species directly affect the growth of the sugar beet. Another genus with high abundance was *Nitrososphaera*, an archaea involved in the Feammox process of Fe^3+^ reduction and ammonia oxidation, a process important for nitrogen cycling and plant growth. This microorganism is a common inhabitant of soils treated with synthetic fertilizers [43], which could explain its high abundance in the rhizosphere of the sugar beet. In the phyllosphere, as in the rhizosphere, Actinobacteriota and Proteobacteriota were the two dominant phyla. As far as we know, the phyllosphere of the sugar beet has not yet been extensively studied. Thompson et al. [38] defined *Pseudomonas* and *Erwinia* as two dominant genera in leaves of the sugar beet collected in different seasons. *Pseudomonas* was also abundant in our samples, but *Erwinia* could not be detected. The most abundant phyllosphere genus detected in the present study was *Lactobacillus*, a group of probiotic bacteria that have also been detected in the phyllosphere of raw eaten produce [44]. This genus has not previously been associated with the sugar beet, but Barbu et al. [45] have shown that *Lactobacilli*, when sprayed on sugar beet leaves, can successfully colonize the phyllosphere. A common species detected in the phyllosphere by metabarcoding was *Komagataeibacter saccharivorans*, a phyllosphere bacterium that can synthesize acetic acid and cellulose, but the role of this bacterium in the plant phyllosphere has not been studied.

To further explore cultivable plant-beneficial species inhabiting the sugar beet, rhizosphere and phyllosphere samples were applied to different media and examined for plant-beneficial properties. The greatest species diversity was obtained on the REa medium containing the sugar beet root extract. This medium has previously been described as effective in isolating bacteria from the rhizosphere of cactus and succulent plants [4], where representatives of *Enterobacter*, *Klebsiella*, *Bacillus*, and *Azospirillum* were isolated. Similarly, we detected several *Bacillus* isolates on this medium, as well as *Pseudomonas*, *Microbacterium*, and *Acinetobacter*. We also identified several plant-beneficial isolates on this medium, mostly from the genus *Bacillus*. Although the highest number of rhizosphere species was isolated on REa, the highest number of plant-beneficial strains was isolated on the 0.01 TSA medium. This medium contains dextrose, trypticase peptone (a pancreatic digest of casein of an animal origin), and phytone peptone (an animal-origin free enzymatic digest of soy) as C and N sources, but in low concentrations. Genera detected on this medium included *Stenotrophomonas*, *Enterobacter*, *Brucella*, *Bacillus*, *Pseudomonas*, and *Acinetobacter*. Most isolates exhibited many co-occurring plant-beneficial traits, especially germination stimulation, siderophore production, and biocontrol potential. This medium has been successfully used to isolate plant growth-promoting bacteria from various plants [46,47,48], but the exact reason for its efficiency in enriching such strains remains to be elucidated. As for the phyllosphere, the highest number of species was obtained on NA, a commercial culture medium. However, the highest abundance of strains with plant-beneficial properties was obtained on LEa, a plant-based medium enriched with the sugar beet leaf extract. In that regard, the rich culture media can be used for sugar beet community studies. However, plant-based media can deliver the best when the goal is to isolate many strains with plant-beneficial properties.

The highest abundance of plant-beneficial strains was found in both the rhizosphere and phyllosphere samples in July, including two species with 8 out of 10 tested plant-beneficial traits: *B. australimaris* and *Enterobacter ludwigii*. *B. australimaris* is a rhizosphere isolate obtained on SEph, a medium enriched with the soil extract and phytagel, an agar substitute produced by bacterial fermentation as a firming agent [49]. This isolate can stimulate germination, produce siderophores, and solubilize phosphate. These properties have previously been associated with *B. australimaris*, and it has been shown that this bacterium can promote the growth of the medicinal plant *Barleria lupulina* [50]. In addition, we demonstrated that *B. australimaris* was active against all sugar beet pathogens tested. Two isolates of *E. ludwigii* were obtained on the 0.01 TSA and SEa mediums (TSA differs from SEph in that it contains agar as a solidifying agent). These two isolates were obtained from the rhizosphere in July but had different characteristics. Both can stimulate germination, produce siderophores, and are active against all pathogens tested. One isolate can solubilize phosphate and the other can produce exopolysaccharides. The plant-promoting properties of this species have been reported in several studies. Solubilization of calcium triphosphate and activity against *Fusarium solani* were associated with *E. ludwigii* isolated from the rhizosphere of *Lolium perenne*. Furthermore, this strain stimulated the formation of lateral roots of *L. perenne* [51]. Additionally, this species has been associated with the promotion of tomato growth, and the tested strain was also active against *F. oxysporum* [52], similar to the isolates in our study. In addition to *E. ludwigii* and *B. australimaris*, several isolates of *S. maltophilia* were detected on different media in July. These isolates occurred in both the phyllosphere and the rhizosphere, and their common feature was activity against sugar beet pathogens. This bacterium is known for its interactions with plants, and its growth-promoting activity has been previously demonstrated for several crops, especially under stress conditions [53,54]. In May, *Acinetobacter caloacetic* isolated from the rhizosphere on the 0.01 TSA medium was the isolate with the highest number of co-occurring plant-beneficial traits. This bacterium is also known to promote plant growth, for example, in *Lemna minor* and *Lactuca sativa* [55]. In October, *Pantoea ananatis* showed the most plant-beneficial traits and was obtained from the rhizosphere on a MSEM medium enriched with the soil extract. This isolate exhibited all tested properties except HCN production and showed activity against several sugar beet pathogens. This species is known for its phytopathogenic properties [56], but there are also reports of its plant growth-promoting activity [57]. In October, two *Bacillus* species (*pumilus* and *safensis*) were also associated with plant-beneficial traits—they could produce siderophores and dissolve phosphate. In other studies, strains of these two species have been shown to produce a wide range of phytohormones and other plant growth-promoting substances [58,59].

Additional experiments are required to confirm our findings. Our dataset’s total number of effective sequences was 2.072 M, and the total number of species observed was 179, which is lower than in previous studies. More plant-beneficial species would possibly be detected with better coverage of the 16S rRNA gene. We sequenced the V3-V4 hypervariable region of the 16S rRNA gene, but the V2 region may provide a higher resolution for lower-ranking taxa. Therefore, some plant-beneficial genera were identified only by the culture-dependent method. These include *Brucella*, *Enterobacter*, and *Paenarthrobacter*. To our knowledge, only *Enterobacter* has been previously associated with the rhizobiome of the sugar beet, while the other two genera were not detected by metabarcoding. Amongst all the detected plant-beneficial genera, the following have been reported as growth promotors for the sugar beet: *Acinetobacter*, *Stenotrophomonas*, *Enterobacter*, *Pseudomonas*, *Bacillus*, *Nocardioides*, and *Pantoea*. Sugar beet-associated genera newly identified as plant-beneficial by the present study are *Achromobacter*, *Brucella*, *Pseudoxantomonas*, *Paenarthrobacter*, *Peribacillus*, and *Paenibacillus*. Therefore, our collection contains several isolates that can be used as biofertilizers and biocontrol agents in sugar beet production after testing its activity in planta. These will be the direction of our future research.

## 5. Conclusions

In summary, culture-dependent and culture-independent techniques are required to obtain a more comprehensive overview of a plant’s microbiome and the diversity of associated plant-beneficial microorganisms. The combination of both methods in this study show that classical metabarcoding with hypervariable regions of the 16S rRNA gene is not sensitive enough to detect strains with plant-beneficial traits and should not be used as a guide for isolation efforts. A culture-dependent microbiome analysis on plant-based media can enable the discovery of plant-beneficial taxa that may not be detected by a 16S rRNA amplicon analysis.

## Figures and Tables

**Figure 1 microorganisms-11-01538-f001:**
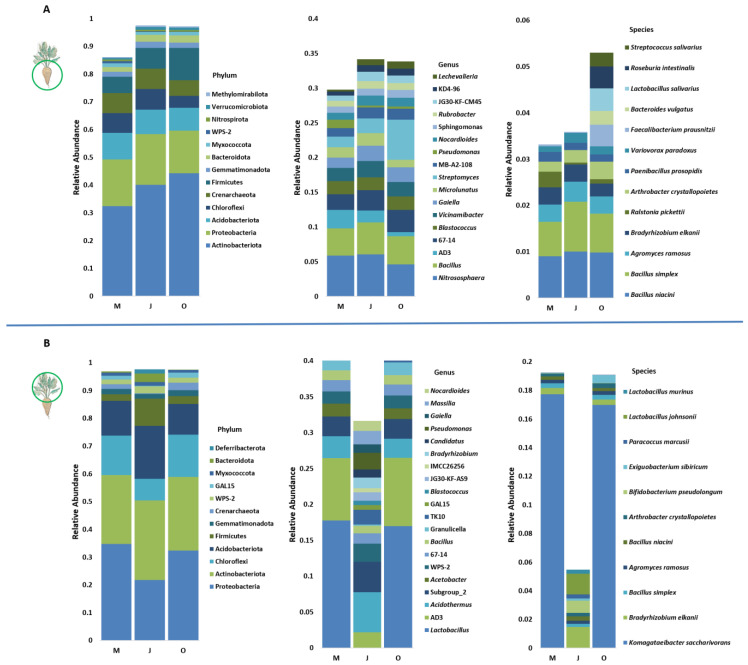
Relative abundance of operative taxonomic units (OTUs) in the microbiome of *B. vulgaris* obtained by 16S rRNA gene metabarcoding in the rhizosphere (**A**) and the phyllosphere (**B**). Only the top taxa with the highest abundance are shown for each sample at 3 different taxonomic levels. OTUs with an abundance greater than 0.05 are shown for phylum, 0.01 for the genus, and 0.02 for species. Different sampling months are indicated with M (May), J (July), and O (October). OTUs denoted with a combination of letters and numbers are candidate taxa that have not been cultured in laboratory conditions.

**Figure 2 microorganisms-11-01538-f002:**
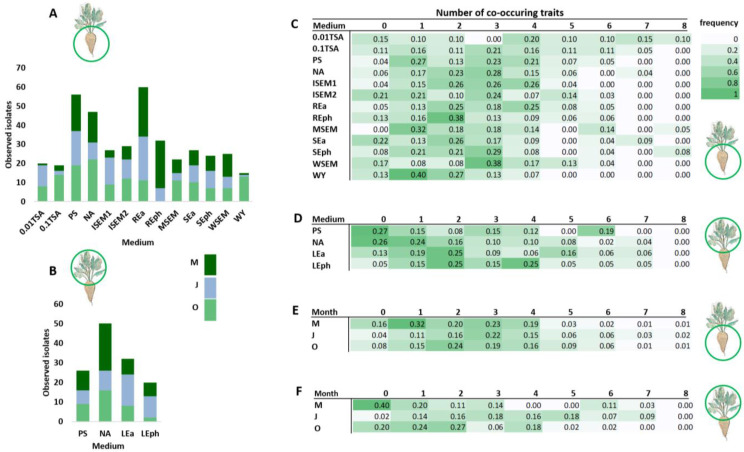
Comparison of different media regarding their effectiveness in isolating strains with plant-beneficial traits. Shown is the number of species isolated on each medium from the root (**A**) or leaves (**B**) of the sugar beet collected in May (M), July (J), or October (O). The frequency of isolates showing a certain number of co-occurring traits was calculated from the total number of isolates obtained on different growth media (**C**,**D**) or in different months (**E**,**F**). The numbers 0 to 8 in (**C**–**F**) represent the number of co-occurring traits—for example, the number 5 represents all isolates with any combination of 5 out of 8 tested traits. TSA, PS, NA, ISEM, REa, REph, MSEM, SEa, WSEM, WY, LEa, and LEph represent different media used for strain isolation (please see Materials and Methods for a more detailed description of these media).

**Figure 3 microorganisms-11-01538-f003:**
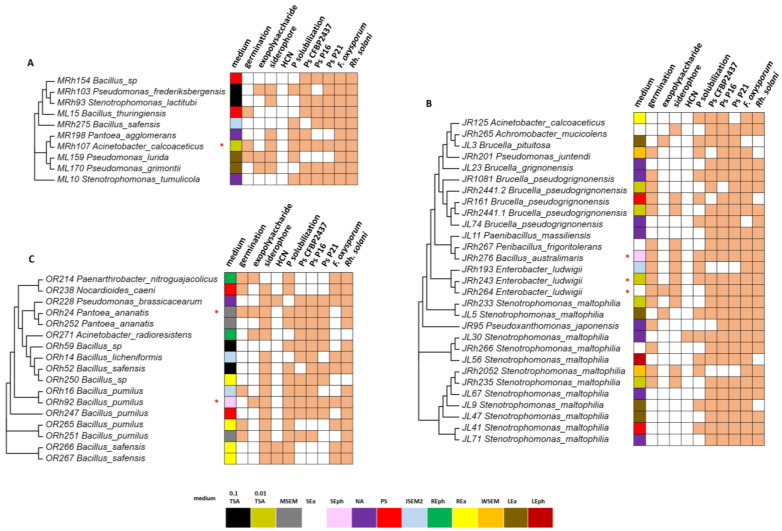
Isolates with five or more co-occurring plant-beneficial traits obtained in May (**A**), July (**B**), or October (**C**). Isolates labeled M (May), J (July), and O (October) represent different months they were sampled, while R represents rhizosphere and L indicates phyllosphere isolates. The different media on which the isolates were obtained are color-coded. A colored square indicates the presence of a plant-promoting trait. * Indicates isolates that have eight co-occurring traits.

**Figure 4 microorganisms-11-01538-f004:**
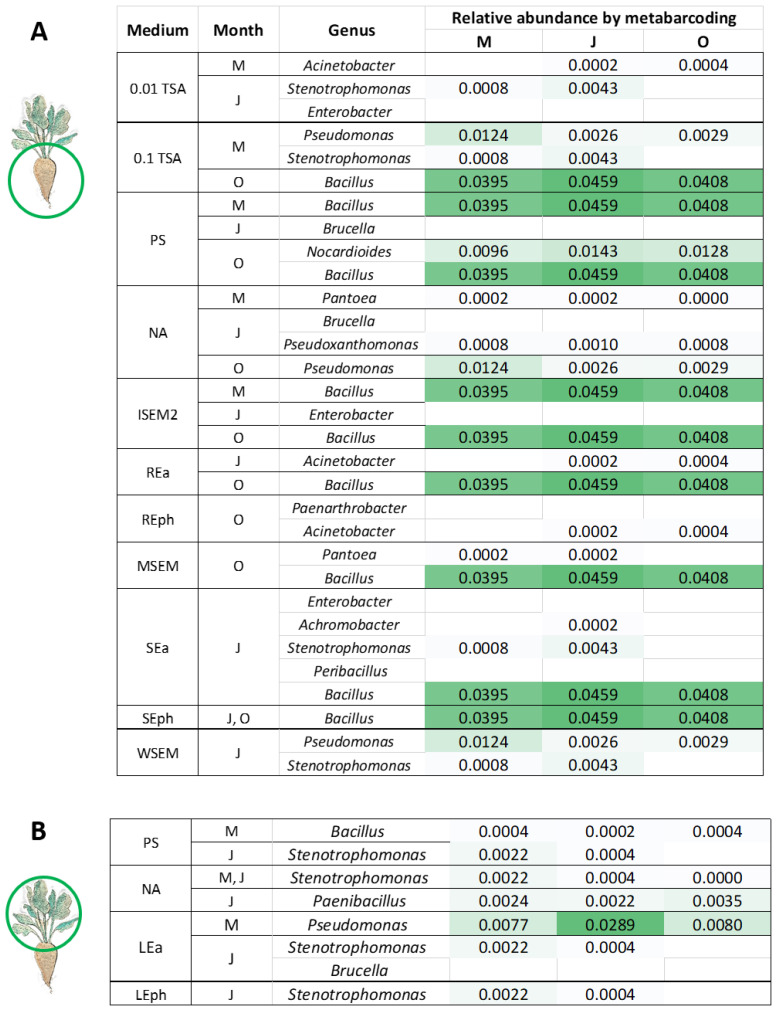
Comparison of the relative frequency of genera determined by metabarcoding with the diversity of genera found on different media. The first column contains media successfully used to isolate plant-beneficial isolates from the rhizosphere (**A**) and phyllosphere (**B**). The Genus column lists only those genera that contain isolates with five or more plant-beneficial traits, as determined in Figure 3. Each genus is paired with the medium on which it was detected and the month it was sampled (M—May, J—July, O—October). The heat map shows the relative abundance of each genus as determined by 16S rRNA metabarcoding.

## Data Availability

Not applicable.

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
