# Peer review of "Culture-Dependent and Metabarcoding Characterization of the Sugar Beet (Beta vulgaris L.) Microbiome for High-Yield Isolation of Bacteria with Plant Growth-Promoting Traits"

_microorganisms, 2023, doi:10.3390/microorganisms11061538_

Round 1

Reviewer 1 Report (New Reviewer)

Minor editing of the English language required

Author Response

Reviewer 2 Report (New Reviewer)

The manuscript was easy to understand, the data of this manuscript were supportive and the conclusion was credible. The illustrations, tables and figures in this manuscript are useful and necessary. The objective is clear and the references are mostly relevant. I agree for its publication after some minor revision listed as following:

1. Line 257-259, Sequencing of the 16S rRNA gene amplicon was performed for molecular identification of bacterial isolates using the universal primers UN1-16SF (GAGAGTTTGATCCTGGC) and UN1-16SR (AGGAGGTGATCCAGCCG). Please give the starting position of the universal primers.

Author Response

Reviewer 3 Report (New Reviewer)

Brief Summary

The manuscript microorganisms-2392846 investigated the suitability of 16S rRNA gene metabarcoding for the efficient isolation of plant growth-promoting bacteria from sugar beet. The study was carried out with valid methodologies. The quality of data handling is good. However, there are many flaws in the quality of the manuscript preparation and the research question being addressed. Presented in this form the manuscript is not useful to advance the knowledge in the field.

First, 16s rRNA gene sequencing could not be proposed as a tool to predict PGPB. It is a molecular approach that responds to different questions than culturable approach in a study of a microbial community. The data retrieved from 16 rRNA sequencing might serve as a basis to study the potentialities of metagenome to express some functions. However, authors did not use this type of bioinformatic analysis or studied the dataset beyond a taxonomic assignment. Statistical or artificial intelligence methods were also not used to compare findings. Second, if proposed as a new approach the authors should have carried out the same evaluation on many different species, also considering extrinsic and intrinsic variables.

The data collected should be exploited in a study of sugar beet microbial community through cultural and molecular methods and considering the seasonal dynamics. The selection of bacteria useful in sustainable agriculture should be considered without linking them in a unique approach with metabarcoding.

Below some other specific comments.

Specific comments

Abstract: The abstract must be shortened and structured with the relevant findings obtained.

Introduction: The introduction correctly places the study in context, considers a good number of update references. Clear statements of the purpose of the study were provided.

·         Add clear working hypotheses.

·         Add a clear aim of the study.

Materials and Methods: The authors described the methods used clearly.

Results and discussion: The results description is clear and supported by appropriate figures.

Discussion: Discussion should be rearranged following the new research questions.

Conclusions: The section should be rearranged following the new research questions.

Other comments

2023 references should also be considered.

The English language needs to be improved.

Round 2

Reviewer 3 Report (New Reviewer)

The authors improved the quality of the English language and smoothed the old research question. However, the wrong concept of using 16s rRNA sequencing as a tool to predict the presence of valuable strain is still present, e.g.:

Lines 107-110 - Despite its proven superiority over culture-dependent methods, we hypothesized that the isolation of bacteria onto specialized media could prove to be equally sensitive or even more sensitive when the goal is detecting and isolating bacteria with special characteristics.

The 16s rRNA sequencing could not be considered an option when the goal is detecting and isolating bacteria with special characteristics.

The concept is reapeted in the other sections, i.e.:

Lines 120-122 - The goal was to compare the lists of taxa obtained by both methods and to determine whether 16S metabarcoding gives insight into the presence of bacteria with several co-occurring plant-beneficial characteristics.

Lines 474-475 - We wondered if that dataset alone would be informative enough to predict the most known plant-associated bacteria with beneficial traits.

The ability of PGP is strain-specific and 16s rRNA sequencing allows the microbial community to be characterized mostly at the genus level. How could your research question be based on the comparison of two techniques that answer different questions?

As I stated in the previous review, as long as the manuscript is addressed in this way, it cannot be considered valid for publication.

The English language steel needs improvements.

This manuscript is a resubmission of an earlier submission. The following is a list of the peer review reports and author responses from that submission.

Round 1

Reviewer 1 Report

The paper by Tomić et al. investigates plant-beneficial bacterial and the microbiome found in sugar beet rhizo- and phyllosphere. A very noble cause, yet the execution of this analysis is lacking. My main concern are the paragraphs 2.2 and 2.4. The authors mention briefly the procedure of obtaining DNA for NGS amplicon sequencing. First of all, the term rhizosphere does not include roots, it is the soil that is adjacent to the roots. The roots contain the microbial microhabitats of rhizoplane and the endorhizome. Were the roots taken with adjacent soil? Were they somehow homogenized/crushed? How were the samples “slightly crushed”? It says further that the samples were shaken in PBS and then filtered (?) – was the DNA extracted from the filters? In my experience the sample suspension would have clogged the filter immediately. The described procedure is very confusing. Even more concerning is the procedure for the bacteria extraction for cultivation. Why was the leaf crushed before sterilization? Fractures in the tissue allow disinfectant entry which kills all the endophytes. The epiphyte extraction seems even more deleterious. Why grind the whole leaf for this? And then “the same procedure as for endophytes was repeated” – the sterilization procedure?! If these procedures were done as described here the majority of data obtained afterwards is worthless. The conclusion in line 570-575 that a lower number of species was found than usually might be proof that the bacteria/DNA extraction procedures were flawed. The strain picking procedure seems also inadequate. The authors admit that the strains picked from all those various media were able to grow on undiluted TSA plates. I find that hard to believe that no strains were lost during this recultivation.  This issues need to be addressed in order to verify the accuracy of the findings.

Reviewer 2 Report

1. Figure 3. phylogenetic tree: The names of some species are cut off.

2. In the first part of the abstract, the purpose of the study should be expressed more specifically.

For example, was 16S rRNA gene metabarcoding first applied to sugar bet?

3. Research objectives should not be placed in the middle of the abstract. It's hard for readers to read.

"The aim was to find the optimal conditions for isolating plant-beneficial bacteria from the sugar beet microbiome and to compare the diversity of bacteria isolated on each media with the phylogenetic diversity described by metabarcoding."

4. The abstract writing is not systematic. The results of the study and the abstract do not match.

The abstract is generally written in the order of the background, purpose, main experimental results, conclusion, and meaning of the study.

5. There is no need to mention the parts that need not be mentioned in the abstract. 

("Actinobacteriota and Proteobacteria are the two dominant phyla in all samples.")

("The highest abundance of plant-beneficial bacteria was found on 0.01 TSA medium from rhizosphere samples collected in July.")

6. line 239: PDA means the 1 x potato dextrose agar?, 

Is the culture medium ingredient listed, and is the name of the culture medium company and country of origin listed?

8. Which is right one?

line 104: Sampling was done three times in 2021, three (May), six (July), and nine (October) months

Line 261: Samples of the rhizosphere and phyllosphere of sugar beet were taken after 3, 5, and 8 months of growth in May, July, and October.

9.line 281-287: Do not repeat the data in the table in the text. There is no reason to repeat.

The writing of the paper is too disorganized. Professional correction is needed.

10. line 190: There is no need to write this down. Readers already know it.

 "If there were two or more morphologically the same colonies from the same medium, only one of them was left while the others were not analyzed further."

11. line 285-295: Instead of lengthy descriptions of what is already shown in Figure 1, number the proportions on the graph for the main classification in Figure 1. Please express your findings briefly.

12. Conclusion generally does not include references. 

If you want to compare your research results with references, discuss it in the 'result and discussion' part.

13. Where did you get plant pathogen? At the Biological Resources center? What is accession number?